# Does microRNA Perturbation Control the Mechanisms Linking Obesity and Diabetes? Implications for Cardiovascular Risk

**DOI:** 10.3390/ijms22010143

**Published:** 2020-12-25

**Authors:** Lucia La Sala, Maurizio Crestani, Silvia Garavelli, Paola de Candia, Antonio E. Pontiroli

**Affiliations:** 1Laboratory of Cardiovascular and Dysmetabolic Disease, IRCCS MultiMedica, 20138 Milan, Italy; paola.decandia@multimedica.it; 2Dipartimento di Scienze Farmacologiche e Biomolecolari, Università degli Studi di Milano, 20133 Milan, Italy; maurizio.crestani@unimi.it; 3Laboratorio di Immunologia, Istituto per l’Endocrinologia e l’Oncologia Sperimentale, Consiglio Nazionale delle Ricerche (IEOS-CNR), 80131 Napoli, Italy; silvia.garavelli@multimedica.it; 4Dipartimento di Scienze della Salute, Università degli Studi di Milano, 20142 Milan, Italy; antonio.pontiroli@unimi.it

**Keywords:** obesity, diabetes, T2D, CVD, atherosclerosis, inflammation, microRNA, metabolic syndrome, cardiovascular complications

## Abstract

Metabolic disorders such as obesity and type 2 diabetes (T2D) are considered the major risk factors for the development of cardiovascular diseases (CVD). Although the pathological mechanisms underlying the mutual development of obesity and T2D are difficult to define, a better understanding of the molecular aspects is of utmost importance to identify novel therapeutic targets. Recently, a class of non-coding RNAs, called microRNAs (miRNAs), are emerging as key modulators of metabolic abnormalities. There is increasing evidence supporting the role of intra- and extracellular miRNAs as determinants of the crosstalk between adipose tissues, liver, skeletal muscle and other organs, triggering the paracrine communication among different tissues. miRNAs may be considered as risk factors for CVD due to their correlation with cardiovascular events, and in particular, may be related to the most prominent risk factors. In this review, we describe the associations observed between miRNAs expression levels and the most common cardiovascular risk factors. Furthermore, we sought to depict the molecular aspect of the interplay between obesity and diabetes, investigating the role of microRNAs in the interorgan crosstalk. Finally, we discussed the fascinating hypothesis of the loss of protective factors, such as antioxidant defense systems regulated by such miRNAs.

## 1. Introduction

Nowadays, obesity is defined as a problem widely diffused in the world: the World Health Organization (WHO) data estimated that at least 1.1 billion people are either overweight or obese (https://www.who.int). Obesity represents the major risk factor for the development of metabolic diseases, such as impaired glucose tolerance (IGT, or prediabetes) and type 2 diabetes (T2D), and predisposes to the increase of cardiovascular (CV) risk over a long-time exposure [1,2]. Obesity also has been recognized as an independent predictor of left ventricular diastolic dysfunction (LVDF) [3], coronary heart disease (CHD), and premature death [4,5]. Recent data from the Framingham study showed that the risk for heart failure (HF) doubled in individuals with a body mass index (BMI) > 30 kg/m^2^ when compared to those with normal BMI [3].

The causes of increased obesity could be identified in the defective and excessive fat quantity and its distribution, metabolic function and specific visceral fat depots, in addition to the sedentary lifestyle and unhealthy habits. However, recent findings demonstrated that visceral adipose tissues (VAT) associates with adverse changes of the common CVD risk factors more than BMI alone; higher VAT is detrimental for cardiometabolic risk factors profile [6,7].

T2D prevalence is increasing worldwide, and it is expected to affect 700 million people by 2045 (https://www.diabetesatlas.org; International Diabetes Federation. IDF Diabetes Atlas—9th Edition); T2D closely associated with obesity epidemics, being the latter one of the major causes of insulin resistance (IR), a key component in the etiology of diabetes [8]. In T2D, the obesity-associated IR is accompanied by endothelial dysfunction (ED), which in turn, is a predisposing factor to atherosclerosis, suggesting that the impact of chronic hyperglycemia may induce an exacerbation of the already existing vascular damage, mainly attributable to the accumulation of lipid-products and inflammatory components in the endothelium [9].

The interplay of obesity, insulin resistance, chronic inflammation, development of T2D and cardiac diseases has been recently highlighted [10,11]. The risk of developing T2D in the obesogenic state increased 3- to 6-fold [11] and dramatically potentiate the already elevated risk of fatal and non-fatal cardiovascular events. Thus, obesity may be considered as a catalyst for more severe diabetes complications. As reported in the Insulin Resistance Atherosclerosis Study (IRAS), subjects with diabetes showed higher atherosclerosis in the internal carotid artery than the normoglycemic subjects [12].

The involvement of the inflammatory components also has an important role in the progression from obesity to diabetes that may affect the endothelium, among other cell types. In the Atherosclerosis Risk in Communities Study (ARIC), both obesity and diabetes exhibit inflammatory activity mediated by adiponectin, although this topic presents conflicting aspects; on one hand, adiponectin may play the role of the protective factor for CVD and increase insulin sensitivity, but on the other hand, its endothelial activation may interfere with the production of nitric oxide (NO), inducing coronary plaque stabilization, arterial vasodilation [9] and finally promoting atherosclerosis [13,14].

In addition to the CV risk highlighted in the aforementioned studies, other predisposing factors for the progression from obesity to T2D (more specifically for IR and glucose intolerance) may be counted in β-cell loss and dysfunction through genetic or β-cell cytotoxic factors. It is well known that the onset of T2D requires biochemical alterations in insulin-sensitive tissues (liver, the skeletal muscle, and the adipose depots), triggering glucose intolerance that finally results in T2D [15]. Hence, β-cells may not be able to fully compensate for decreased insulin sensitivity.

The secretion of non-esterified fatty acids (NEFAs) by adipose tissue in obese people may lead to the hypothesis that IR and β-cell dysfunction are most likely linked [16]. Nevertheless, although the role of proinflammatory cytokines has emerged in pancreatic β-cell dysfunction and T2D pathogenesis, it remains still unclear whether fat-derived factors can destabilize and disrupt β-cell compensation and lead to its dysfunction. T2D pathogenesis involves the communication between different metabolic tissues triggering β-cell mass progressive dysfunction. In particular, adipose tissue plays a central role in the regulation of the metabolism of the surrounding organs via cytokines and NEFA secretion [17].

Many studies have focused on the influence of different hormones and circulating factors on the inflammatory component, such as the synthesis and release of adipokines, chemokines, growth factors and other proteins. A recent study shows the ability of the adipose tissue surrounding the pancreas, called peripancreatic adipose tissues (*P*-WAT), in secreting inflammatory factors in a rat model of diabetes [18], suggesting interorgan crosstalk. Indeed, *P*-WAT secreted factors have been shown to increase β-cell proliferation in a rat model of diet-induced obesity [19].

It is plausible that the metabolic state and nutritional requirements would be modulated in response to epigenetic processes, such as chromatin arrangements for stimulating structural adaptations that control gene expression. However, several post-transcriptional modulators may control the mechanistic link between obesity and diabetes, such as miRNAs, histone deacetylases (HDACs) and histone acetyltransferase (HATs), but the two latter are not included in this review.

Recent advances have identified microRNAs—a class of highly conserved non-coding RNAs of 25 nucleotides in length able to modulate gene expression—as key regulators of gene expression, directly repressing or secondarily activating genes involved in the metabolic reprogramming and glucose homeostasis. Although several miRNAs have been well recognized as a modulator of patho-metabolic state in both T2D and obesity [20,21,22], the relationship connecting these two intermingled conditions is still not completely clear.

## 2. miRNAs Landscape and General Characteristics

In this review, we sought to underline the involvement of miRNAs in metabolic regulation, which might open up new clinical and therapeutic frontiers in the prevention and treatment of obesity, diabetes and related pathologies.

A novel class of small non-coding RNA (miRNAs) of 18–26 nucleotides in length was identified as key controllers of gene expression. MiRNAs are transcribed from genomic DNA into a primary transcript, called pri-miRNA, which is larger than a mature miRNA [23]. The pri-miRNA undergoes cleavage by the endonuclease DroshaRNase III and generates an intermediate of around 60–70 nucleotides known as pre-miRNA, which is transported into the cytoplasm by the Exportin-5/Ran-GTP complex [24,25,26] and processed by Dicer to form a mature miRNA [27]. At this stage, miRNAs can regulate gene expression at the post-transcriptional level by targeting the 3′ untranslated region (3′-UTR) of mRNA transcripts. MicroRNAs can be located within coding DNA sequences (exons/introns, and across a splice site) or in intergenic regions. The different miRNA location influences the transcription modality: intronic miRNAs likely use their own promoter, while miRNAs in intergenic regions are considered evolutionarily more conserved because they use the host gene transcriptional machinery and could be transcribed with their host gene, sharing common regulatory patterns [28].

miRNAs are also present in body fluids, such as blood, urine and cerebrospinal fluid [29], showing stability dictated possibly by their association with either vesicular structures as the extracellular vesicles (EVs) or associated with RNA-binding proteins (Argonaute2 [Ago2]) or lipoprotein complexes (high-density lipoprotein [HDL]) [30].

Furthermore, recent work has demonstrated that miRNAs, being exported in the extracellular space in association with the EVs, may be up-taken by recipient cells and by multiple mechanisms such as endocytosis, micropinocytosis and phagocytosis and influence their functions. Because of their stability in the circulation, miRNAs are currently explored for their potential as biomarkers for cardiovascular disease (CVD). Thus far, distinctive patterns of circulating miRNAs have been described for atherosclerotic disease [31], type 2 diabetes mellitus (DM) [32] and have been shown to describe the progression from prediabetes to diabetes [33,34] and hypertension, among other diseases. Not least, recently, EVs are also emerging as potential novel therapeutic interventions for metabolic diseases, i.e., endogenous miRNAs were able to be encapsulated in engineered EVs, allowing them to reach the target tissues and exert a beneficial metabolic response [35].

miRNAs discovery approaches have been used to detect miRNAs expressed in many diseases such as cardiovascular, metabolic, as well as in cancer and other, and under specific cellular conditions in which the miRNAs contributing is considerable time-consuming, but provided the basis in finding their role in disease-disease relationships and understanding miRNA-based therapeutics. Although many technical advancements have also been done with the use of machine learning systems, most of these approaches are still in their infancy. Meta-analysis studies on miRNA expression have revealed that the majority of differentially expressed miRNAs in diabetes are downregulated, presumably as a result of DNA-methylation (DNA-me) mechanisms affecting miRNA loci.

## 3. miRNAs and Cardiovascular Risk Factors (CRFs)

The assessment of subjects with modifiable risk factors (RF) is of paramount importance for delay or blocking disease the progression of the disease. Among the RFs, those cardiovascular (CV) are of increasing concern worldwide. The CRFs are multifactorial, and the accumulation of several RFs exponentially increases the risk of cardiovascular events. Targeting a specific CRF would reduce CV risk, but addressing simultaneously multiple risk factors may synergistically reduce even further the risk of a CV event.

It is speculated that several miRNAs are considered as risk factors for CVD due to their correlation with cardiovascular events. Many miRNAs have been associated with conventional CRFs as hypertension, dyslipidemia, obesity and diabetes (which substantially increase the risks for cardiovascular complications), and thus, miRs may be targeted to reduce CV risk.

Concerning the associations of miRNAs with hyperglycemia, a canonical and modifiable risk factor for obesity and diabetes, the circulating miR-21 stands out. It has been linked with endothelial damage induced by ROS, lipid peroxidation and antioxidant defense inhibition, in the DIAPASON study [32], in which prediabetes and diabetes state was diagnosed with 2-h plasma glucose (2hPG) test, as recommended by American Diabetes Association (ADA) guidelines [36]. In addition, circulating miR-15a has been linked with retinal damage in the same aforementioned group of patients [37]. In a cohort of 52 Iranian subjects categorized as diabetics or prediabetics based on the fasting plasma glucose (FPG) and hemoglobin A1c (HbA1c), miR-30d-5p and miR-126-3p were associated with prediabetic state [38]. In the PREDAPS study [39], serum microvesicles miR-10b and miR-223-3p were identified as able to discriminate the non-progressing prediabetics from those progressing towards diabetes. Interestingly, the miRNA target analysis showed their associated pathways, such as insulin signaling and response to inflammatory stimuli [40].

Another CRF is hypertension, which is highly prevalent in obesity and diabetes [41]. Among miRs related to hypertension, miR-145 may be associated with the pathogenesis of pulmonary arterial hypertension [42]. In rat models, spontaneous hypertension was characterized by an upregulation of miR-145 expression in the thoracic aorta compared with the control group [43]. The authors have identified solute carrier family 7 member 1 (SLC7A1) as a direct miR-145 target, with possible therapeutic implications. In hypertensive patients, lower levels of nitric oxide (NO), linked with endothelial dysfunction, have been associated with miR-122, which binds the 3′UTR of SLC7A1 and induces the reduction of SLC7A1 levels [44]. Of note, SLC7A1 is a transporter gene for L-arginine and normal NO metabolism [45]. In a cross-sectional study exploring the molecular mechanisms of white coat hypertension (WCH), miR-155 could have a predictive value being associated with blood pressure values and inflammatory markers [46]. Also, in endothelial cells, the increase of miR-155 expression is inversely related to eNOS expression and NO release, suggesting a crucial role in endothelium-dependent vasorelaxation [47]. In-vivo genetic evidence showed that a pleiotropically acting miR-31 and its target protein phosphatase 6c (ppp6C) are critical intrinsic factors for controlling physiological and pathological immune responses regulated during hypertension [48]. Also, miR-431-5p is considered as a potential key regulator in angiotensin-II-induced vascular injury in mice made hypertensive by 14-day of angiotensin-II infusion, being correlated with blood pressure [49]. Mounting shreds of evidence showed that several miRs are associated with the components of the renin-angiotensin-aldosterone system (RAAS), the latter highly activated in the etiology of hypertension [50]. The renin expression seems to be controlled by many miRs, e.g., miR-181a-5p and miR-663 in hypertensive individuals [51].

Relevant to the link between obesity and diabetes, miR-26a has been associated with a higher risk for the development of coronary artery disease (CAD). The latter miR seems to control critical pathways, such as BMP/SMAD1 signaling, and target genes relevant to endothelial cell growth, angiogenesis, and left ventricular (LF) function post-myocardial infarction (MI) [52]. In addition, in subjects with coronary heart disease (CHD), plasmatic miR-125b could be used as a diagnostic value of the occurrence of stenotic lesions [53].

Clear pieces of evidence are represented by the involvement of miRNAs in modulating genes responsible for glucose-homeostasis in obesity-induced T2D: 220 miRNAs differentially expressed were identified in pancreatic islets, adipose tissue, and liver of diabetes-resistant and diabetes-susceptible mice [54]. In particular, the overexpression of the liver miR-143 has emerged in the impairing AKT activity, which seems to be important in leading to unbalanced glucose homeostasis and to the development of IR. Thus, miR-143 may be considered as a therapeutic target for the treatment of obesity-associated diabetes [55].

However, miRNAs exert their biological effect not only at the tissue-specific level but also at the level of peripheral blood. Several studies in humans and rodents supported their role as biomarkers for diabetes and metabolic syndromes due to their increase during glucose intolerance (miR-24, miR-30d, miR-34a, miR-126, miR-146 and miR-148a), disease progression (miR-122, miR-133, miR-210), β-cell injury (miR-375) and inflammation (miR-21-5p) [56,57,58]. In addition, distinct circulating miRNA profiles between subjects with obesity and T2D aroused considerable attention given their potential as biomarkers for these diseases. Indeed, strong correlations were found between some circulating miRNAs and the development of metabolic syndrome [59], obesity [60,61], diabetes [32,62,63] and/or linked to CVD [37,64]. Nowadays, miRNAs are attractive candidates as innovative biomarkers and have been proposed to promote the progression to diabetes [29,33,62]. Furthermore, in bariatric metabolic surgery (BMS), miRs might be effective for making the decision about surgery and/or predicting weight loss after BMS [65], providing targets for future treatments.

## 4. The Pathological Roles of miRNAs in Obesity and Diabetes: Pancreatic Islets and Adipose Tissue

Several miRNAs correlate with islet damage. MiR-375 deserves special attention; it was the first resident miR to be identified as pancreas-specific, evolutionarily conserved and islet-specific miRNA involved in modulating glucose-induced insulin secretion in pancreatic islets. Considered as a β cell-specific biomarker, miR-375 is sensitive and reliable for monitoring islet cell damage, i.e., the overexpression of miR-375 suppressed its validated target gene, myotrophin (MTPN) deputed to insulin-secretion [66]. A further aspect urging us to investigate miR-375 concerns the islet amyloid formations in β-cells, whose aggregation causes islet β-cells deficit [67]. However, β-cells replication could also be modulated by miR-7a through its modulation of mTOR signaling, which promotes adult β-cells replication in mouse primary islets, electing miR-7a as a target for diabetes treatment [68].

To discern the role of miRNAs associated with insulin sensitivity, He et al. performed a miRNA microarray analysis of skeletal muscles from either healthy or T2D rats, in which the major miR involved in insulin-stimulated glucose uptake belonged to the miR-29 family: its upregulation in diabetic animals was able to control insulin response into adipose tissue, muscle and liver. Interestingly, the miR-29 level was upregulated in hyperglycemia and in hyperinsulinemia in 3T3-L1 adipocytes [69]. Performing a study to investigate whether microRNAs or DNA methylation contributes to β-cell-specific silencing of MCT1, Pullen et al. showed that specific members of the miR-29 family (miR-29/a and miR-29/b) were highly expressed in mice islets. They also demonstrated that inhibition of miR-29a in primary mouse islets increased MCT1 mRNA levels, showing the contribution to the β-cell-specific silencing of the MCT1 transporter affecting insulin release [70]. This evidence is confirmed by the findings that miR-29a/b knockout mice have impaired insulin exocytosis and show alterations in peripheral tissue insulin sensitivity resulting in impaired glucose tolerance [71].

Another miRNA involved in metabolic and diabetic disease onset is miR-15b; its upregulation has been observed in the regenerating mouse pancreas through the regulation of NGN3, a bHLH transcription factor essential for insulin expression [72,73,74].

Adipose tissue (AT) has well been identified as an important endocrine whole-body energetic regulator due to its feature to release hormones and to regulate glucose and lipid homeostasis [75]; at the same time, AT represents an important source of extracellular miRNAs, showing that AT is able to exert paracrine communications between the liver, skeletal muscle, pancreas and cardiovascular system (Figure 1). In a study performed on adipose-specific DICER1 knockout mice, a dramatic decrease of miRNA levels both in the blood circulating EVs was reported.

It has been suggested that obesity-dysregulated microRNAs in exosomes derived by visceral adipocytes might be used to predict the impairment of insulin receptor signaling in obese subjects.

Many miRs showed pro-adipogenic features (Table 1 and Table 2): the miR-103 and miR-143 were identified as relevant molecules both in vitro and in vivo studies [76,77], and miR-200 increased lipid accumulation and expression of fatty acid-binding protein-4 (FABP4) in ST2 marrow stromal cells [78], the miR-17-92 cluster was upregulated 2–3-fold during the early clonal expansion stage of 3T3-L1 adipogenesis. The latter study also identified Rb2/p130 as a final target of miR-17-92 [79]. However, the modulatory activity of miRNAs, such as miR-let-7, miR-27 and mIR-138, specifically hampers adipogenesis [80,81,82,83].

Exo-miRs, such as miR-122, miR-192, miR-27a-3p and miR-27b-3p, can also inhibit the expression of PPAR-α in WAT. However, miR-27a seems to facilitate the crosstalk between adipocytes and skeletal muscle, inducing IR through the inhibition of PPARG [84]. MiR-27a is critical for obesity by regulating IR in adipocytes, as well as acting as a repressor of adipocyte differentiation and facilitating inflammation in adipose tissues in virtue of its role in regulating M1-like macrophage polarization [85]. It is acknowledged that obesity is closely related to the resident cells of AT, such as macrophages, and of its metabolic state [86] (Table 1 and Table 2).

In obese mice, macrophages were shown to secrete a higher amount of the proinflammatory exo-miR-155 than lean mice [87], reducing hypertrophy and aging of 3T3-L1 adipocytes [88] through the inhibition of PPAR-γ gene expression. This protein reduces insulin sensitivity in the liver and causes glucose intolerance and IR after injection in control mice.

Mesenchymal stem cell-derived exo-miR-124a can silence forkhead box A2 (FOXA2) in macrophages, leading to intracellular lipid accumulation [89]. In addition, the plasma exosomal transcription release factor (PTRF) increases the occurrence of hypertrophy and aging of 3 T3-L1 adipocytes. Circulating polymerase I and PTRF, like adipokines, may partially contribute to the deleterious effects of visceral fat accumulation [90]. Notably, adipocyte-derived exosomes are essential for liver physiological activity [91].

Transplanting white and brown adipose tissue caused a restoration of miRNA levels and an improved glucose tolerance due to a decrease in circulating FGF21, which is modulated by several miRNAs, including miR-99b [92]. Adipose tissue is also responsible for the release of vesicles, which activate macrophages and induce expression of TNF-α and IL-6, causing IR by activating TLR4 [93]. Povero et al. demonstrated the crucial role of hepatic stellate cells (HSCs) activation in liver fibrosis observed in chronic diseases such as NAFLD, and that is mediated by internalized miR-128-3p, miR-122 and miR-192 (contained in EVs released by lipid-induced HepG2 cells, or hepatocytes) [94,95]. This issue confirmed the role of miRNAs as a strong link between diabetes and obesity; indeed, increased levels of circulating miR-122 in human subjects are associated with IR, obesity and diabetes [96,97].

Another evidence that further highlights the role of miRNAs in regulating adipogenesis is given by miR-130b, highly expressed in adipocytes. MiR-130b is then secreted by 3T3-L1 cell line during adipogenesis [98], and microvesicles overexpressing this miRNA showed a downregulation of PPARγ that blocked adipogenesis in vitro [99]. Exosomes can also be involved in maintaining pancreatic β-cell mass by inducing their proliferation in vitro through an enrichment in miRNA-16, which in turn regulates genes, such as PTCH1, involved in pancreatic development [100,101] (Table 1 and Table 2).

## 5. The Crosstalk Mediated by Exo-miRNAs in Obesity and Diabetes

Increasing evidence on the ability of miRNA to provide relevant biological information being passed among cells and tissues support the fascinating hypothesis of miRNAs as endocrine and paracrine messengers.

Many extracellular miRNAs are shuttled by membrane-bound extracellular vesicles (EVs) and could exert cell-to-cell communication in physiological and pathological situations. In this regard, some microRNAs have been extensively studied in the crosstalk between peripheral tissues, especially in adipose tissue, and in the endocrine pancreas, such as in β-cell failure during metabolic disorders like T2D [116]. In this view, under the influence of the surrounding organs, microRNA patterns may suffer some changes affecting the levels of significant proteins, driving the dysregulation of β-cells.

Indeed, recent studies showed that EVs produced in the endosomal compartment of cells transfer mature miRNAs from adipose tissue to surrounding organs [92], which can affect, in some cases, whole-body insulin sensitivity [88]. In this context, adipose exosomal miRNAs (exo-miRs) are emerging as regulators of gene expression in distant tissues [92].

In an animal study, the influence of high-fat diet (HFD) modifies plasma exo-miRs pattern, reporting an increase of exo-miR-192 and exo-miR-122 linked with IR [117], miR-27a-3p and miR-27b-3p. Transfected exosomes with mimic miR-192, miR-122 and miR-27a/b induced some changes in the activation of adipocyte-PPAR/RXR pathway, as well as in the network of lipid metabolism, inducing central obesity and hepatic steatosis in recipient mice [117].

A recent study in humans, in spite of the low number of the subjects enrolled, showed a peculiar signature of exo-miR expression, in a cohort of obese people, with or without T2D (respectively, Ob/D^+^ and Ob/D^−^) [118]: 15 Ob/D^−^-specific miRNAs were identified, one-third of which belongs to the let-7 family and is related to vascular damage, atherosclerosis and β-cell stress; whereas 9 miRNAs are Ob/D^+^-specific (miR-150-3p, -let7b-5p, -664a-5p, -320a, -320b, -423-5p, -424-3p, -483-5p, -2960), affecting the pathways associated with insulin secretion, insulin regulation and energy metabolism. The highest expressed exo-miR found in Ob/D^-^ vs. Ob/D^+^ with a potential link to the progression from obesity to diabetes was miR-23a-5p [118].

Human islets transplanted in nude mice (a mouse model of xenoislet transplantation) releases exosomes containing specific microRNAs into the circulation: Vallabhajosyla et al. suggested that transplant tissue-specific exosomes can be purified from recipient plasma [119] using anti-HLA antibody-conjugated beads to provide a non-invasive way to track the state of transplantation. This approach has the potential to improve diagnostic accuracy compared with whole plasma analysis of exosomes. Of interest, exosome miRNA-cargo exhibits four microRNAs commonly expressed at the highest level in transplanted tissue-specific exosomes and in its islet graft: miR-191-5p, miR-23a-3p, miR-16-5p, and miR-24-3p. As shown in Figure 1, a kind of resident cells in pancreatic islets such as T-lymphocytes release vesicles enriched in miR-142-3p, miR-142-5p and miR-155, inducing apoptosis targeting Cxcl10 [120] and expression of cytokines Ccl2 and Ccl7 [113].

With regard to exosome-based crosstalk within the cardiovascular system, a mention of exosome influence on atherosclerotic burden is mandatory. It has been shown that under shear stress, endothelial cells release EVs with increased levels of miR-143 and miR-145, which are internalized by smooth muscle cells (SMCs) of tunica media to avoid SMC dedifferentiation, and they hence exert a protective role against the atherosclerotic plaque formation [114]. However, miRNAs directly hit target tissue, such as miR-15a, which has already been shown to play an important role in insulin production in pancreatic β-cells also has been found to be increased in plasma of diabetic subjects. After being delivered by exosomes and internalized in retinal cells, it may play a role in retinal injury, targeting AKT3 [115].

## 6. The Mechanism Related with miRs and the Antioxidant Systems in Obesity

Obesity damages mitochondrial function, which could be one of many defects linking obesity to diabetes, by decreasing insulin sensitivity and compromising β-cell. Therefore, the unbalanced components of reactive oxygen species (ROS) appear critical for the development of diabetic CVDs.

The fascinating hypothesis of the lack of protective factors, such as antioxidants defense [9,32,121], might explain some of the abnormalities typical of metabolic derangements. There is growing literature that suggests the prominent roles of mitochondrial dysfunction and/or oxidative stress in defects of glucose metabolism. A role for oxidative stress in the development of T2D has been largely supported by several studies [32,121,122,123,124,125]; however, oxidative damage is also associated with obesity, can also be prevented by overexpression of CuZn- or Mn-superoxide dismutase (SOD1 and SOD2, respectively) [126]. In this regard, the role of miR-21, notoriously upregulated in human obesity [127,128], and diabetic milieu (including its cardiovascular complications [129,130,131,132,133,134,135,136,137]), in reducing the plasmatic amount of SOD2 was recently established, highlighting the issue of ROS-mediated damage in subjects overweight and with a high risk of developing diabetes [32]; besides, a high amount of lipid peroxidation was found in these subjects compared to normal conditions. This issue has been recently corroborated in a cellular model of glucose variability (GV), in which overexpression of miR-21 presumably induces negative modulation of antioxidant response in endothelial cells [138]. The increasing of miR-21 in adipose tissue of obese subjects with T2D suggests a link between this miR and the onset of insulin-resistance within of pathophysiologic progression from obesity to T2D [107].

Obesity has been correlated with low levels of glutathione (GSH); also, diabetic milieu, as well as the fluctuations of the glucose from euglycemic to hyperglycemic range, exhibits a reduction of mRNA levels, activity, and protein expression of glutathione peroxidase-1 (GPx-1) [121]; this suggests a common soil for the onset of metabolic abnormalities. In this regard, high levels of miR-185 were implicated in the inhibition of GPx-1. MiR-185 has an important role in fatty-acid metabolism regulation, cholesterol homeostasis in hepatocytes through the regulation of sterol regulatory element-binding proteins (SREBP-2), and insulin sensitivity [109]. The control on SREBP-2, a sensor for the synthesis and absorption of cholesterol able to induce the receptors for the low-density lipoprotein (LDL) [108], is transposed on the repression of SREBP-2, causing a reduction of LDL uptake and HMG-CoA reductase activity. MiR-185 also regulates the accumulation of lipids and enhances the insulin signaling pathway by upregulating the insulin-receptor substrate-2 (IRS2).

## 7. miR-Based Pharmacological Intervention

RNA-interference technologies, based on siRNA and similarly on miRNAs, are promising therapies for many diseases, but the low specificity of miRNAs, the high degree of redundancy and the multiple miRNA-targets hampered the development of a specific pharmacological treatment and of diagnostic biomarkers. Despite these limitations, preclinical studies and clinical trials have been proposed, in particular for the use of miRNA mimetics and anti-miRNA oligonucleotides. Thanks to their low toxicity, anti-miR-103 and anti-miR-107 (called RG-125/AZD4076 developed by Regulus Therapeutics) completed phase I clinical trials and are ongoing in phase II for the treatment of T2D, or prediabetes, with nonalcoholic fatty liver disease (NAFLD) and nonalcoholic steatohepatitis (NASH) (http://clinicaltrials.gov—NCT02826525 and NCT02612662), by targeting insulin signaling in adipose tissue, in the liver and reducing adipocyte size. It seems that miR-103 and miR-107, increased in obese mice, by targeting caveolin-1 (CAV1) may regulate glucose homeostasis [102]. Since it was estimated the high prevalence of NAFLD and NASH in populations suffering from obesity, the treatment with anti-miR-103/107 may be promising. Other applications of miR-based therapeutics regard engineered delivery vesicles, such as EVs, loaded with specific (or patterns) of miRNAs (or anti-miRNAs) to counteract/activate specific pathways.

## 8. Conclusions

The dysregulation of microRNA patterns, intra and/or extracellular, may be considered on the basis of metabolic abnormalities registered in obesity and diabetes. Above all, the changes in miRNA patterns in EVs are well described in both obese and diabetic patients, suggesting that EVs can carry other miRNAs that may participate in the development of their CV complications. A better understanding of the miRNA-based crosstalk among different tissues may help the design of miR-based pharmacological approaches to regulate and restore systemic metabolism.

## Figures and Tables

**Figure 1 ijms-22-00143-f001:**
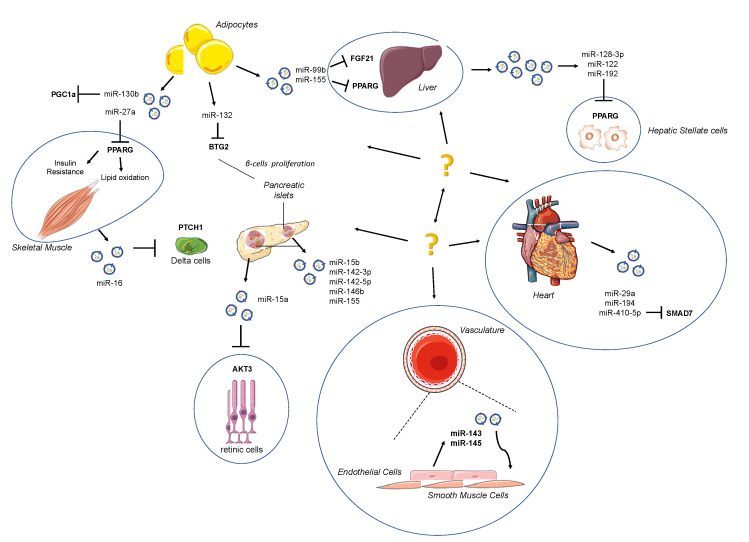
Schematic depiction of exo-miRs released by surrounding organs. Adipose tissue (AT) acts as an active endocrine and energy-supply organ. It has emerged as a source of extracellular microRNAs (miRNAs), exerting a paracrine communication between the liver, skeletal muscle, pancreas and cardiovascular system. The strong connection among organs may be attractive targets for novel therapeutic strategies.

**Table 1 ijms-22-00143-t001:** Mammalian miRNAs regulating adipogenesis and insulin sensitivity.

miRNA	Source System	Effect	Targets	Ref.
miR-103	3T3-L1	Pro-adipogenic	ERK5, MAPK7	[76]
miR-143	3T3-L1	Pro-adipogenic		[76,77]
miR-200	ST-2	Pro-adipogenic		[78]
miR-17-92	3T3-L1	Pro-adipogenic	Rb2/p130	[79]
let-7	3T3-L1	Anti-adipogenic	HMGA2	[80]
miR-27	3T3-L1, OP-9	Anti-adipogenic		[81]
miR-138	MSCs	Anti-adiopogenic	EID-1/Indirectly PPARγ	[82,83]
miR-29	3T3-L1	Inhibitor of glucose uptake	INSIG1 CAV2	[69]
miR-15b	Murine pancreas	Pancreas regeneration, insulin expression	NGN3	[72,73,74]
miR-29 a/b	MIN6 β-cell line	β-cells identity	MCT1	[70]
miR-375	Islets	Insulin secretion in β-cells	MTPN	[66]
miR-103 miR-107	Obese mice liver	Insulin sensitivity		[102]
miR-34a miR-210 miR-383	Obese T2D model	β-cells apoptosis and/or glucose-induced insulin-secretion inhibition		[103,104,105]
miR-127-3p miR-184		Insulin secretion in β-cells		[106]
miR-7a	Murine and human pancreatic islets	Negatively affects β-cells proliferation	mTOR	[68]
miR-21	Adipose tissue of obese subjects	IR		[107]
miR-185-5p	Human HepG2 hepatocytes	Cholesterol biosynthesis LDL uptake Lipid metabolism Insulin sensitivity		[108,109]
miR-146a	Mice with Diet-induced metabolic disease	Repress inflammation and diet-induced obesity Regulates metabolic processes		[110]
miR-192 miR-193b		Adipocyte differentiation Lipid homeostasis		[111,112]

**Table 2 ijms-22-00143-t002:** Exosomal miRNAs as mediators of interorgan crosstalk in diabetes and obesity.

microRNA	Releasing Tissue	Target	Reference
miR-99b	Adipose Tissue	FGF21 (liver)	[92]
miR-155	Adipose Tissue	PPARγ (liver)	[88]
miR-128-3p miR-122 miR-192	HepG2 hepatocytes	PPARγ (hepatic stellate cells)	[94]
miR-130b	3T3-L1	PPARγ (adipocytes)	[99]
miR-16	Mice skeletal muscles	PTCH1 (pancreas)	[100,101]
miR-142-3p/-5p miR-155	Pancreatic Islets T cells	Ccl2 Ccl7 Cxcl10	[113]
miR-143 miR-145	Endothelial cells	SMCs	[114]
miR-15a	Pancreas	AKT3 (retinal cells)	[115]

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
