# Peer review of "Does microRNA Perturbation Control the Mechanisms Linking Obesity and Diabetes? Implications for Cardiovascular Risk"

_ijms, 2020, doi:10.3390/ijms22010143_

Round 1
Reviewer 1 Report
In this paper, the authors summarized the miRNAs roles in obesity and diabetes, also introduced the role of miRNAs in cross-talk between organs, as well as the association with cardiovascular risk factors. But there are a few points should be addressed:
- The arrangement of this paper is not easy for read, I recommended to rearrange the structure as:
1) introduce and miRNa biogenesis
2) miRnas as biomarkers for obesity and diabetes;
3) The pathological roles of miRNAs in obesity and diabetes( which subtitle with pancreatic, adipose….)
4) The crosstalk mediated by exo-miRNAs in obesity and diabetes
5) The mechanism related with miRNA and obesity(antioxidant, mitochondria, …)
…
- The introduction mentioned inflammation are related with obesity and diabetes, but this was not be discussed in the text.
The title shows this paper should emphasis on the cardiovascular diseases, but the manuscript seems more general discuss about all organs, only small part about cardiovascular pathologicial mechanism and risk factors, which is not enough.
Author Response
Dear Reviewer 1,
we appreciated your precious comments and recommendation.
Regards your points, we restructrured as:
introduction, we preferred alone in order to explain the general picture of the problem; whereas mir biogenesis we would keep separate from the intro because it could be too long.
we have changed the paragraph title as recommended.
the inflammation is mentioned in the introduction as a part fundamental to understand the context of these pathologies. We decided to not insert it in total part in the review, because the review could be considered extremely long and complicated to read. The inflammation is a chapter of the obesity and diabetes that, in our opinion, it would be need to extensively considered as a separated paper.
Reviewer 2 Report
In this manuscript of La Sala et al, the authors have done a pretty good job in reviewing the current literature about a research Topic that is recently capturing a growing attention. In fact, the potential role of microRNA as both diagnostic biomarkers and therapeutic targets, is a fascinating aspect of pharmacological research and deserves particular attention especially for pathologies such as cancer, obesity and diabetes. In this review, the authors report the microRNAs involved in the regulation of pathological mechanisms underlying obesity and diabetes and their implication for cardiovascular risk.
My general opinion is that the review is well organized and therefore I have only few minor suggestions:
Line 39: Please define what do you mean for general population.
Lines 49 and 57: Please revise the verb tense.
From line 123 and 125: Please, include one or more references showing the role of EVs as microRNA carriers and their involvement in the cell to cell communication including the neuro-immune axis. Perhaps, you could add animal studies too.
Line 335: What do you mean for pancreatic β-cell mass?
Table 1: Personally I would change the title "Model System" with "Source".
Table 2: Personally I would add the column "Effect" also in this table.
Author Response
Dear Reviewer 2,
thank you very much for your precious and punctual comments and suggestions.
in line 39 general population: we preferred to eliminate this term "general population" to avoid misunderstanding.
line 49-57: we revised the verb tense.
line 335: beta cell mass is the total quantity of beta cells, separated by islets.
table 1. we added your suggestions in yellow.
table 2. We appreciated your suggestions to add a column "effect", but as reported in the ahead of the table, the effect is referred to cross-talk.